# A Novel Strategy for the Production of Edible Insects: Effect of Dietary Perilla Seed Supplementation on Nutritional Composition, Growth Performance, Lipid Metabolism, and Δ6 Desaturase Gene Expression of Sago Palm Weevil (*Rhynchophorus ferrugineus*) Larvae

**DOI:** 10.3390/foods11142036

**Published:** 2022-07-09

**Authors:** Khanittha Chinarak, Worawan Panpipat, Atikorn Panya, Natthaporn Phonsatta, Ling-Zhi Cheong, Manat Chaijan

**Affiliations:** 1Food Technology and Innovation Research Center of Excellence, School of Agricultural Technology and Food Industry, Walailak University, Nakhon Si Thammarat 80160, Thailand; khanittha.ch@mail.wu.ac.th (K.C.); cmanat@wu.ac.th (M.C.); 2Food Biotechnology Research Team, Functional Ingredients and Food Innovation Research Group, National Center for Genetic Engineering and Biotechnology (BIOTEC), 113 Thailand Science Park, Phaholyothin Rd., Khlong Nueng, Khlong Luang, Pathumthani 12120, Thailand; atikorn.pan@biotec.or.th (A.P.); nat-thaporn.pho@biotec.or.th (N.P.); 3Zhejiang-Malaysia Joint Research Laboratory for Agricultural Product Processing and Nutrition, College of Food and Pharmaceutical Science, Ningbo University, Ningbo 315211, China; cheonglingzhi@nbu.edu.cn

**Keywords:** sago palm weevil larvae, insect, perilla seed, nutritional value, growth performance, lipid metabolism

## Abstract

The nutritional value, growth performance, and lipid metabolism of sago palm weevil larvae (*Rhynchophorus ferrugineus*, SPWL) raised on plant-based diets (soybean, rice bran, and ground sago palm trunk (GSPT)), supplemented with various concentrations (0, 3, 7, 15, and 20%) of perilla seed (PS) were compared with traditional diets i.e., regular GSPT (control) and GSPT supplemented with pig feed. All supplemented diets rendered SPWL with higher lipid and protein contents (*p* < 0.05). Supplementing with 7–20% PS enhanced α-linoleic acid content in SPWL, resulting in a decrease in the n-6:n-3 ratio to a desirable level. Dietary PS supplementation increased Δ9 (18), total Δ9 and Δ5 + Δ6 desaturase indexes, fatty acid (FA) unsaturation, and the polyunsaturated FA:saturated FA ratio in SPWL, while lowering atherogenicity index, thrombogenicity index, and Δ6 desaturase (*fads2*) gene expression. Boosting with 7% PS improved the majority of growth parameters and enhanced essential amino acid and mineral contents (*p* < 0.05).

## 1. Introduction

Edible insects raised on farms can be regarded as an alternative food source that benefits the environment by emitting less greenhouse gases and ammonia, having greater feed conversion ratios, providing less disease and zoonotic risk, and using substantially less water than traditional livestock farming [1,2]. The use of insects as a food and feed source is thought to have major benefits for the economy, the environment, and food security. Although it is still a very small niche industry in the EU, insect farming for food and feed is growing in popularity [3]. Additionally, there is a challenge with the import of insects and insect-derived products into the EU for food and feed because the use of insects is more common outside the EU [3]. Considering the full supply chain, from farming to consumption, a risk profile and presentation of potential biological, chemical, allergenicity, and environmental concerns connected with farmed insects used as food and feed was developed [3].

Because it is a source of lipids, proteins, and minerals, the sago palm weevil larvae (SPWL) is a popular farm-raised edible insect that is widely commercialized and promoted by the government extension sector in Southern Thailand [4,5,6]. When fed a commercial animal feed designed with a normal ground sago palm trunk (GSPT) diet, the SPWL includes a considerable quantity of lipid, about 60% of its dry body weight, and is primarily constituted of palmitic acid and oleic acid [4]. Consequently, there has been a lot of discussion on the effect of diets on the nutrient composition, growth performance, and gene expression of insects [7]. SPWL’s fatty acid (FA) composition could be manipulated by modifying the feed composition, notably increasing n-3 essential FA, to lower the n-6:n-3 ratio and so improve its health effects [5,6]. An imbalance in the n-6:n-3 ratio has been linked to health problems. The ratio of n-3 to n-6 polyunsaturated fatty acid (PUFA) should be less than 5 for optimal human health [8].

According to previous research, increasing the amount of α-linolenic acid (ALA) and essential amino acids (EAA), while decreasing the n-6:n-3 ratio by incorporating perilla (*Perilla frutescens*) seed (PS) into a regular GSPT diet, improved the nutritional values of SPWL [4]. PS is one of the botanical sources of n-3 FA, notably ALA, and crude protein, and it could be employed as a feed additive to boost ALA levels in animals [8,9]. According to Peiretti et al. [10], increasing PS inclusion increased the concentration of PUFA in rabbit muscle and perirenal fat while decreasing the concentrations of saturated fatty acid (SFA), monounsaturated fatty acid (MUFA), and the n-6:n-3 PUFA ratio. In addition, Deng et al. [9] found that adding PS to the diet raised the contents of ALA, eicosapentaenoic acid (EPA), docosapentaenoic acid (DPA), and vaccenic acid while decreasing the n-6:n-3 ratio in lamb muscle and liver. Ducks fed a basal diet supplemented with 5% PS had significantly higher protein and ALA concentrations in their breast and liver [11]. Through a series of enzymatic reactions, ALA can be transformed into EPA and docosahexaenoic acid (DHA), but n-6, linoleic acid, can also be elongated by the same enzymes [12]. Several studies have reported the inclusion of plant-based ALA in animal feed and have shown that, following insect ingestion, ALA can be an efficient strategy to produce desired health consequences for individuals [13]. Thus, if insects are to be a component of long-term and healthy foods, they should have relatively high quantities of n-3 PUFA, resulting in a more favorable n-6:n-3 ratio. Even when fed the identical diet, changes in accumulation efficiency and de novo FA biosynthesis among species are likely to result in diverse FA profiles in various species. To obtain the best diet formula, it is necessary to address the proper PS level in SPWL’s diet.

While PS supplementation enhanced PUFA levels in SPWL, it tended to impede growth performance and lower SPWL survival rate in long-term feeding [5,6]. To find the best SPWL diet formula, a mixture of the selected plant-based ingredients, including soybean meal, rice bran, and GSPT, was prepared [5], in combination with PS. The objective of this study was to investigate the effect of various PS concentrations in a formulated diet on growth performance, nutritional value, and the activities of lipid metabolism enzymes, including the expression of Δ6 desaturase (*fads2*) genes in SPWL. The Δ6 desaturase enzyme, encoded by the *fads2* gene, is one of two rate-limiting enzymes that convert the PUFA precursors-ALA (n-3) and linoleic acid (n-6) to their respective metabolites, by catalyzing the addition of a double bond to fatty acids at the sixth carbon–carbon bond position from the carboxylic acid end [14,15,16] (Figure 1). The *fads2* has 12 exons and 11 introns, and 444 amino acids are encoded by its conventional transcripts, which begin at 1335 base pairs [14].

This is the first study on the use of dietary supplementation from perilla seeds as a cutting-edge method for creating edible SPWL. The new findings could help increase the commercial value of this local insect for future food security and sustainability.

## 2. Materials and Methods

### 2.1. Diets and SPWL Rearing

To compare the effects of feed ingredients on nutrient composition, growth performance, lipid metabolism indices, and Δ6 desaturase (*fads2*) gene expression in SPWL, commercial pig feed (PF), mixed plant-based ingredients (PI, a 1:1 mixture of soybean and rice bran), and PI formulated with different levels of commercial PS (Bankongloi, Chiang Mai, Thailand) (0, 3, 7, 15, 20%, *w*/*w*) were supplemented to the basal GSPT (control). The supplemented diets were made by combining GSPT with each supplement in a 2:1 (*w*/*w*) ratio and used to rear SPWL in three separate lots. Table 1 lists the ingredients in the experimental diets. All the diets were analysed for their proximate composition (see Section 2.3) and FA composition (see Section 2.4) in comparison with the control diet.

Experimental diet nomenclature: C: control, PF: diet supplemented with pig feed, P0–P20: diets contained 0–20% perilla seed, respectively.

According to Chinarak et al. [5], the SPWL were reared at room temperature (27–30 °C) with a relative humidity of 70–80%. The Walailak University Animal Ethics Committee has evaluated and approved this rearing procedure (WU-AICUC-64029). In brief, 4 mature weevil pairs were grown in a spherical plastic container with 3 kg of individual feed meal and 1 L of water. The SPWL were formed after 20 days of rearing and fixed at 120 larvae/container with a total rearing period of 40 days. To avoid a feed shortage, new diets (1.5 kg each time) were added on days 20 and 30 of rearing. The SPWL were measured on days 20 and 40 to assess growth performance. The live larvae reared on day 40 were rinsed with tap water, blanched, and subjected to freeze-drying before being kept in a plastic box at −20 °C until further investigation. The nutritional composition, lipid metabolism indices, and gene expression of Δ6 desaturase (*fads2*) were also determined in freeze-dried SPWL.

### 2.2. Growth Performance

Twenty larvae from each group were used to determine biometric characteristics. The larvae were individually measured for body weight (g) and total length (cm) at day 40 after rearing. SPWL biometric parameters were calculated using the following formulae:(1)Dry matter content =(dry weight / live weight)×100
(2)$$\mathrm{Condition}\;\mathrm{factor}=\left(\mathrm{body}\;\mathrm{mass}\;\times 100\right)\;/{\;\mathrm{body}\;\mathrm{length}}^{3}\;$$
(3)Survival=(numbers of final larvae / numbers of initial larvae)×100
(4)Growth rate (g/day)=(lnfinal body weight−lninitial body weight) / day 

### 2.3. Proximate Composition and Mineral Profile

The proximate composition of the SPWL, including moisture, crude protein, crude fat, and ash, was investigated [17]. To avoid overestimation of protein content due to the presence of some non-protein nitrogen in SPWL, a conversion factor (CF) of 6.25 was used for the diets, but a CF of 5.6 was used for the SPWL [4,5]. Carbohydrate was calculated by subtracting 100 from the moisture, fat, protein, and ash contents. Inductive couple plasma (ICP) spectrometry was used to evaluate the elemental composition of SPWL, which contained potassium (K), phosphorus (P), sodium (Na), magnesium (Mg), calcium (Ca), zinc (Zn), manganese (Mn), iron (Fe), and copper (Cu) [17].

### 2.4. FA Composition

The FA composition of feed and SPWL was determined using a gas chromatography/quadrupole time of flight (GC/Q-TOF) mass spectrometer, as described by Chinarak et al. [5].

To assess lipid quality, the n3/n6 ratio, total SFA, total unsaturated fatty acids (UFA), total MUFA, and total PUFA were calculated. The ratio of PUFA to SFA, n-6 to n-3 PUFA ratio, atherogenicity (IA) index, and thrombogenicity index (IT) of the SPWL lipid were calculated as follows [5]:(5)$$\mathrm{IA}=\frac{C12:0+4\times \;C14:0+\;C16:0}{\Sigma \mathrm{MUFA}\;+\;\Sigma \mathrm{PUFA}\;\omega -6+\;\Sigma \mathrm{PUFA}\;\omega -3}$$
(6)$$IT=\frac{C14:0+\;C16:0+\;C18:0}{0.5\Sigma \mathrm{MUFA}\;+0.5\Sigma \mathrm{PUFA}\omega \;-6+3\Sigma \mathrm{PUFA}\;\omega \;-3+\frac{\mathrm{PUFA}\;\omega -3}{\mathrm{PUFA}\;\omega -6}}$$

### 2.5. FA Metabolism Indices and fads2 Gene Expression of SPWL

The activities of desaturases in converting SFA to MUFA were measured by dividing the proportion of product by the percentage of precursor [18]:(7)$$\Delta 9-\mathrm{desaturase}\;\left(18\right)\;\mathrm{index}:\Delta 9-\mathrm{DI}\;\left(18\right)=100\;\left[C18:1\;/\;\left(C18:1+C18:0\right)\right]$$
(8)$$\Delta 9-\mathrm{desaturase}\;\left(16\right)\;\mathrm{index}:\;\Delta 9-\mathrm{DI}\;\left(16\right)=100\;[C16:1\;/\;\left(C16:1+C16:0\right)$$
(9)$$\mathrm{Total}\;\Delta 9-\mathrm{DI}\;=100\;\left[\left(C16:1+\;C18:1\right)\;/\;\left(C16:1+\;C16:0+\;C18:1+C18:0\right)\right]$$

The 5 + 6 desaturase index (DI) measures the efficacy with which essential FA such as LA and ALA are converted to longer chain PUFA, whereas the thioesterase and elongase indices assess the extent to which myristic acid (C14:0) is converted to palmitic acid (C16:0) and then to steric acid (C18:0) [18].
Δ5 + Δ6-DI = 100 [C20:2ω-6 + C20:4ω-6 + EPA + C22:5ω-3 + DHA / C18:2ω-6 (LA) + ALA + C20:2ω-6 + C20:4ω-6 + EPA + C22:5ω-3 + DHA](10)
(11)$$\mathrm{Elongase}\;\mathrm{index}\;\left(\mathrm{EI}\right)=\;C18:0/C16:0$$
(12)$$\mathrm{Thioesterase}\;\mathrm{index}\;\left(\mathrm{TI}\right)=\;C16:0/C14:0$$

The expression of Δ6 desaturase (*fads2*) in SPWL was determined using quantitative real-time PCR (qRT-PCR). Degenerate primers of Δ6 desaturase (*fads2*) were designed based on known desaturase sequences from *Danio rerio* (GenBank accession: BC049438.1), *Mus musculus* (GenBank accession: BC057189.1), *Rattus norvegicus* (GenBank accession: BC081776.1), *Argyrosomus regius* (GenBank accession: KC261978.1), and *Nibea mitsukurii* (GenBank accession: GQ996729.1). The β-actin (Forward 5′-GATTCTGGAGATGGT, Reverse 5′-TCTGGGCAACGGAAC) and *fads2* (Forward 5′-TGAACCAGTCRTT GAAG, Reverse 5′- GCTGGATGGYTRCARC) primers were used to quantify the relative gene expression of SPWL.

TRIZOL reagent (Invitrogen, USA) was used to extract total RNA from 40-day raised SPWL using the manufacturer’s protocol. RNA concentration and purification were determined at 260 and 280 nm, respectively. RNA integrity was determined using electrophoresis on a 1% (*w*/*v*) agarose gel. Reverse transcriptase was used to reverse transcribe RNA into first-strand cDNA (iScript^TM^ Select cDNA Synthesis Kit, Bio-Rad, Hercules, CA, USA).

The *fads2* mRNA levels of SPWL were measured using qRT-PCR. For qRT-PCR, a reaction mixture (10 μL) comprised of cDNA (template, 1 μL), forward and reverse primer (10 pmol/μL each, 0.25 μL), 5× Hot FIREPol Evareen qPCR Mix Plus (ROX, 2 μL), and PCR grade water (6.5 μL) was mixed. The denaturation temperature cycle was carried out at 95 °C for 12 min, 95 °C for 30 s, 40 cycles of 55–60 °C for 30 s, and 72 °C for 30 s. The comparative threshold cycle (Ct) method was used to estimate the relative *fads2* gene expression in comparison to β-actin (reference gene).

### 2.6. Cholesterol Content

Cholesterol content of the whole SPWL sample was determined using a gas chromatography-triple quadrupole mass spectrometer (GC/QQQ, GC 7890B/MSD 7000D, Agilent Technologies, Santa Clara, CA, USA) connected to the PAL autosampler system (CTC Analytics AG, Zwingen, Switzerland). Data was acquired by the MassHunter software (Version 10.0, Agilent Technologies). The calibration curves were prepared using cholesterol at different concentrations ranging from 30 to 2000 μg/μL [6].

### 2.7. Amino Acid Profile

The amino acid compositions of SPWL were determined using the Shimadzu-GCMS-TQ8050 NX (Kyoto, Japan). The amino acid content was given as g/100 g dry sample. Furthermore, the essential amino acid index (EAAI) and biological value (BV) were estimated using the formula given below by Chinarak et al. [4]:(13)$$\mathrm{EAAI}=\sqrt[{9}]{\frac{g\;\mathrm{of}\;\mathrm{lysine}\;\mathrm{in}\;100\;g\;\mathrm{of}\;\mathrm{analysis}\;\mathrm{protein}\;\times 100}{g\;\mathrm{of}\;\mathrm{lysine}\;\mathrm{in}\;100\;g\;\mathrm{of}\;\mathrm{reference}\;\mathrm{protein}}\times \left(\mathrm{etc}.\;\mathrm{for}\;\mathrm{other}\;8\;\mathrm{EAA}\right)}$$
(14)BV=1.09×EAAI−11.7

### 2.8. Statistical Analysis

The Statistical Package for the Social Sciences (SPSS) 24 for Windows was used for statistical analysis. Each lot was tested three times. One-way ANOVA was used for statistical analysis. Duncan’s multiple-range test was used to discover significant differences (*p* < 0.05) across samples when comparing means.

## 3. Results and Discussion

### 3.1. Basic Composition and FA Profile of Diets

The effect of PI formulated with various levels of PS (0–20%), then combined into GSPT in comparison to PF-formulated GSPT (2:1, *w*/*w*) and control GSPT alone on the proximate composition of the resulting diets is shown in Table 2. The amount of fat, protein, ash, and carbohydrate varied depending on the type of supplement used (Table 2). Carbohydrate was the most prevalent composition in all diets (63.7–93.9%), due to the high content of starch in the GSPT [4]. The addition of PI, PI plus PS, and PF to GSPT resulted in a significant increase in protein (9.4–10.6 fold) and fat (6.3–22.4 fold) content over the common control diet (*p* < 0.05). The chemical compositions of mixed diets were primarily determined by the major component presented in individual supplements, in which the amounts of soybean, rice bran, PF, and PS formulated for each diet formula were varied (Table 1). Deng et al. [9] found that combining PS with other ingredients resulted in a difference in the composition of lamb diets, particularly protein and fat.

FA composition of all diets is given in Table 3. The most abundant FA in the GSPT were C16:0 (palmitic acid), C18:0 (stearic acid), C18:1 (oleic acid), and C18:2 (linoleic acid), accounting for 62.45 g/100 g of lipid. It should be noted that the FA profiles were found to be similar in all test diets, though their contents differed based on the type of supplement used (Table 1). PS formulation led to a significant increase in C18:3 (all-cis-cis-9,12,15-linolenic acid; ALA), which was 8–26 times greater than the control diet (*p* < 0.05). This led to an increase in n-3 FA enhancement following PS fortification into PI mixed GSPT diets, resulting in PS being an excellent source of ALA [19]. SFA, MUFA, and PUFA differed statistically between the test diets (*p* < 0.05), ranging from 16.87 to 47.26 g/100 g total lipid, 13.90 to 36.55 g/100 g lipid, and 8.96 to 66.94 g/100 g lipid, respectively (Table 1). The n-3 PUFA content of the PI and PS added to the GSPT was primarily increased, leading to a low n-6/n-3 (Table 3).

### 3.2. Growth Performance of SPWL

Table 4 shows the appearances and growth performance of SPWL fed on various test diets. When PI, PS, and PF were combined into GSTP, all growth factors of SPWL were significantly increased when compared to those fed a control diet (*p* < 0.05). The type of supplement and the concentration of PS had an impact on live weight, survival rate, and growth factors. All SPWL raised by 40 days had an 80% survival rate (Table 4). Increased PS levels in PI mixed diets had a negative impact on SPWL survival when compared to the control diet. This result was consistent with the findings of Chinarak et al. [5], who found that incorporating PS into GSPT reduced the survival rate of SPWL. The presence of anti-nutritive substances in PS, such as phytic acid, may have contributed to an imbalance in insect metabolism [5]. The antinutrient found in PS, specifically perilla ketone, a terpenoid composed of furan rings with six carbon chains and ketone functional groups, has been linked to the induction of pulmonary edema and symptoms of perilla mint toxicosis in cattle fed large amounts of PS [20]. Typically, anti-nutritional compounds have a significant negative impact on animal digestive performance [11]. According to Oonincx et al. [8], the addition of 0–4% flax seed oil to the diet of house crickets had a negative impact on survival rate, which decreased from 69% to 55%. SPWL fed PF, PI, and PI formulated with 3–15% PS-formulated GSPT had the highest live weight (*p* < 0.05). There were no significant differences in dry weight of SPWL between supplemented diet groups (*p* > 0.05). SPWL fed supplemental diets had increased dry weight, dry matter content, and condition factor compared to control SPWL (*p* < 0.05). Thus, the addition of supplements into regular GSPT could improve the growth performance of the SPWL. Each treatment had a slight variation in the condition factor. The incorporation of PI and PS into GSPT resulted in a slower growth rate than the PF-diet and control diet (Table 4).

### 3.3. Proximate Composition, Cholesterol Content, and Mineral Profiles of SPWL

Table 5 displays the proximate composition of SPWL fed various diets. The moisture of the SPWL fed with supplemented diets was lower than that of the SPWL fed with the control diet (*p* < 0.05). SPWL fed with the PI and PI with 3% PS-mixed diets had the greatest protein content (*p* < 0.05), which was 20% higher than SPWL fed the PF diet. Compared to other insects, SPWL had a greater protein content than *Tribolium castaneum* (15.3–17.0 g/100), *Imbrasia epimethea* (20.1 g/100 g), and *Imbrasia truncate* (19.1 g/100 g) [21,22]. However, the protein of the SPWL was lower than that of *Allomyrina dichotoma* (54.2 g/100 g), *Tenebrio molitor* (53.2 g/100 g), *Protaetia brevitarsis* (44.2 g/100 g), *Zophobas morio* (46.8 g/100 g), *Gryllus assimilis* (65.52 g/100 g), *Ruspolia differens* (44.3 g/100 g), and *Periplaneta americana* (49.4 g/100 g) [1,23,24]. Lipid was the most abundant component found in SPWL (Table 5), with concentrations ranging from 34.9 to 56.4 g/100 g. The increase in fat content was found in all SPWL reared on all supplemented diets when compared to those fed the control diet (*p* < 0.05). SPWL had a higher lipid content than *Allomyrina dichotoma* (20.2 g/100 g), *Protaetia brevitarsis* (15.4 g/100 g), and *Tenebrio molitor* (34.5 g/100 g) [23]. SPWL had an ash content ranging from 2.8 to 4.2 g/100 g (Table 5). SPWL raised on all supplemental diets had a decrease in ash content when compared to those fed the control diet (*p* < 0.05), but there was no significant difference in the ash content of SPWL raised on the supplemental diet groups (*p* > 0.05). This result indicated that the supplement types had no impact on the inorganic substances of SPWL (Table 5). The amount of ash in edible insects varied according to species and diet [4,5,6]. The carbohydrate content of SPWL fed supplemental diets was lower (18.2–23.1 g/100 g) than that of SPWL fed control diets (38.7 g/100 g) (*p* < 0.05). This was due to the high carbohydrate content of the GSPT in the control diet (Table 2).

The SPWL fed with the PF-formulated diet had the highest cholesterol content, while the SPWL fed with the GSPT control diet showed the lowest cholesterol content (*p* < 0.05). SPWL fed PI-enriched or PI with PS diets (3–20%) had significantly lower cholesterol levels than SPWL fed a PF-fortified diet (*p* < 0.05) (Table 5). This finding was in line with the observations of Batkowska et al. [25], who found that the addition of linseed and soybean oils to hen diets led to a significant reduction in cholesterol content in the yolks. The cholesterol content of SPWL fed with PS-formulated diets differed slightly (Table 5). Cholesterol is a sterol found in animal-based foods that is naturally produced by animal cells [26]. It serves as a substrate for the manufacture of important molecules such as steroid hormones, vitamin D, and bile acids, as well as maintaining cell membrane integrity [27]. A high intake of cholesterol has been connected to an increased risk of hypercholesterolemia, cardiovascular disease, and coronary artery disease [28,29]. The adult population has been advised to ingest less than 300 mg of cholesterol per day [30].

Minerals are required for the maintenance of some life-sustaining physicochemical processes, despite the fact that they do not produce energy, and they play important roles in many bodily activities [31]. Natural minerals found in edible insects include Ca, Cu, Zn, Fe, Mn, Mg, Na, and P [24]. Table 5 shows the macrominerals found in SPWL, which include K, P, Mg, Ca, and Na. K was the most plentiful element in the SPWL (6421–9265 mg/kg) (Table 5). It was greater than the comparative values of conventional meats such as pork (5043 mg/kg), beef (6247 mg/kg), and chicken (5557 mg/kg) [23]. The P content of SPWL ranged from 2926–4273 mg/kg (Table 5), with 3% PS diet-reared SPWL having the greatest P content (*p* < 0.05). SPWL fed PI and PI with PS-supplemented diets had significantly higher P levels than those fed PF-added diet (*p* < 0.05). As a result, SPWL fed PI or PI with PS-containing diets had comparable or even higher P content than those fed the control diet. However, the P content of SPWL fed all diets was enough for a daily requirement of 700–4000 mg/day for good health as informed by the World Health Organization [32]. Ghosh et al. [23] stated that, in contrast to plant-based P, P in insects is readily bioavailable, indicating a potential source for humans. The SPWL fed with the 3% PS diet had the highest Mg content; however, SPWL fed PI or higher PS than 3% added diets had greater Mg content than those fed with the control diet (*p* < 0.05). All SPWL raised on PI or PI with PS-formulated diets had higher Mg levels than those fed the CF diet (*p* < 0.05). Mg content in the SPWL was significantly greater than that reported by Araújo et al. [24] for *Gryllus assimilis* (271 mg/kg) and *Zophobas morio* (391 mg/kg). The Ca content of SPWL ranged from 441 to 699 mg/kg (Table 5), which was greater than that of conventional animal foods such as pork (379 mg/kg), beef (187 mg/kg), and chicken (323 mg/kg), with the exception of chicken eggs (2348 mg/kg) [23]. The Na content of SPWL ranged from 986 to 1351 mg/kg (Table 5), with less Na present in SPWL after all supplemental diet feeding than after control diet feeding (*p* < 0.05). The Na content of SPWL fed only PI or PI-formulated with different PS concentrations differed slightly, but all were higher than that of SPWL fed the PF diet (*p* < 0.05). Na is required for cellular homeostasis and physiological functions; however, excessive Na consumption has been linked to hypertension and non-communicable disease. SPWL raised on a 3% PS added diet had the highest Ca concentration, which was comparable to those fed with a control diet (*p* > 0.05). The feeding of only PI or PI with PS higher than 3% based diets resulted in lower Ca content in the resulting SPWL, but their contents were still higher than that fed on the PF diet (*p* < 0.05). Ca is primarily an essential element, with its phosphate salts playing critical roles in neuromuscular function as well as many enzyme-mediated activities such as blood clotting, bone formation, and tooth formation [33]. The most important micro-minerals in humans are Zn, Mn, Fe, and Cu, which are depicted in Table 5 for current SPWL. The insects are thought to aid in the supply of minerals, particularly Fe and Zn [24], with current SPWL compost containing 81.8–104.6 mg/kg of Zn and 14.2–19.0 mg/kg of Fe, respectively (Table 5). The increased PS level in the diets resulted in lower Zn and Fe contents in the resulting SPWL (*p* < 0.05), except in the SPWL fed a 3% PS diet. Surprisingly, the highest Zn and Fe concentrations were found in SPWL fed on a 3% PS-formulated diet (*p* < 0.05), which could be attributed to the proper absorption of both microelements from the feed ingredients. However, higher PS intake in SPWL fed on high PS-formulated diets may alter mineral absorption from the gut by chelating with phytate, resulting in lower element uptake. The Zn content of these SPWL was higher than that of *Gryllus assimilis* (52.2 mg/kg) and *Zophobas morio* (24.7 mg/kg), with both insects having higher Fe content (27.8 and 22.7 mg/kg, respectively) than the SPWL [19]. Infants, children, teenagers, and women of childbearing age, particularly pregnant women, are the most vulnerable to iron deficiency [33]. Thus, SPWL can be used as an iron alternate. The Mn concentration of SPWL ranged from 10.4 to 49.9 mg/kg (Table 5), with SPWL fed all supplemental-added diets having significantly lower Mn content than those raised on the control diet (*p* < 0.05). This finding may be related to the reduction of Mn availability caused by dietary factors, specifically the lower total concentration of Mn in the experimental diet affected by the dilution of GSPT content in the supplemental-added diets. The Mn in this current SPWL was higher than *Tribolium castaneum* (4.9 mg/kg) [21], but lower than *Allomyrina dichotoma* (86.4 mg/kg), and *Protaetia brevitarsis* (58.9 mg/kg) [23]. The Cu content of SPWL ranged from 10.2–26.1 mg/kg (Table 5). The greatest Cu content was found in the SPWL reared on the PF-formulated diet (*p* < 0.05). SPWL reared on PI and PI withPS-added diets had a significantly lower Cu content than those fed on PF-added diet (*p* < 0.05), though the Cu concentrations were slightly greater or comparable to that reared on the control diet. This could be explained by the PF’s high Cu content, which was consistent with our previous study on the same insect [4]. Despite the fact that SPWL has a lower Cu content, it can be used as an alternative Cu source. As a result, incorporating 3% PS into PI and GSPT could significantly improve the overall element content in SPWL, which could play a significant role in food security. On the other hand, this current study addressed the availability of minerals as influenced by the experimental diet composition, which has not previously been addressed in this insect.

### 3.4. FA Composition and Health Promoting Indices of SPWL

The content and composition of lipids in insects is determined by feed or de novo synthesis, and they are stored as body fat before being degraded, digested, and delivered to their final target cell [34]. Table 6 shows the FA composition of the lipids presented in SPWL as affected by dietary ingredients. The FA composition of the experimental diets, as expected, significantly altered the FA composition of the resulting insect. Increased PS-formulated concentrations in PI-mixed regular GSPT had a significant effect on FA composition, specifically SFA, MUFA, and PUFA of SPWL (Table 6). In SPWL fed with increasing PS levels, there was a progressive decrease in total SFA with increasing PUFA compared to those raised on a control diet (*p* < 0.05). The highest C16:1 content was found in SPWL reared on a 7% PS-added diet (*p* < 0.05), whereas there was no significant difference in C18:1 content between SPWL fed PF-, PI-, and PI with 3–15% PS-formulated diets (*p* > 0.05). Certain ingredients, particularly soybean meal and rice bran, may contribute to the presence of C18:1 in the resulting SPWL, as these are excellent sources of linoleic acid [35]. It should be noted that the SPWL fed with PF-, PI-, and PI with 7% PS-added diets had the highest MUFA content, with no difference among them (*p* > 0.05). The highest C18:2 content was observed in SPWL raised on a PF-diet (*p* < 0.05), whereas feeding SPWL on PI- and PI with 3–20% PS diets resulted in an FA which was 1.4–3 fold lower (Table 6). The SPWL reared on all supplemental diets had a higher C18:2 concentration than those reared on the control diet (*p* < 0.05). This finding indicated that PF and all plant-based ingredients contribute to the C18:2 content of the SPWL [5]. The addition of 3 to 20% PS to the experimental diets increased C18:3 content in the SPWL by 12.40–105 fold and 5.80–49.40 fold, respectively, compared to those fed control and PF-added diets (*p* > 0.05). Notably, adding PF to the GSPT diet did not improve C18:3 content in SPWL compared to feeding GSPT alone (*p* > 0.05), while 3% PS formulation did not significantly increase C18:3 content in SPWL compared to rearing on the PI-added diet (*p* > 0.05). The SPWL fed PF, 7% PS, and control diets had the highest C20:4 content (arachidonic acid, ARA), while those raised on 7% PS-added diets had the highest C20:5 content (EPA), with no significant difference between the other groups (*p* > 0.05). SPWL fed a PI-formulated diet had the highest C22:6 (DHA) content, while SPWL reared on other diets had no significant difference in DHA concentration (*p* < 0.05). When PS inclusion levels in the experimental diets were increased, there was a reduction in total SFA with an increment in PUFA in SPWL, resulting in an increase in the ratio of PUFA to SFA (*p* < 0.05).

The PUFA/SFA ratio is commonly associated with the risk of cardiovascular disease caused by foods with a higher ratio potentially lowering LDL-cholesterol and serum cholesterol levels [36]. This study was similar to the findings of Peiretti et al. [10], who revealed that increasing PS inclusion in diets increased PUFA concentration while decreasing SFA and MUFA contents in the *longissimus dorsi* muscle and perirenal fat. The inclusion of linolenic acid-containing PS in the diets increased the contents of linolenic acid, EPA, and DPA in the *longissimus dorsi* muscle and liver [9]. Furthermore, increasing PS inclusion in the experimental diets resulted in a decrease in total n-6 PUFA content and an increase in total n-3 PUFA content in the SPWL (*p* < 0.05). PS formulations ranging from 3 to 20% in the test diets contributed to a 1.3–67.1 fold decrease in the n-6/n-3 ratio in the SPWL when compared to those fed a PF-diet (*p* < 0.05; Table 6). To determine the nutritional value of lipids for human consumption, the PUFA/SFA and n-6/n-3 ratios are frequently utilised, and a low n-6/n-3 ratio, particularly one of 4:1 or less, is more beneficial in preventing cardiovascular disorders [2,9,25]. The PUFA/SFA ratio of SPWL was significantly increased by feeding 15–20% PS-based diets, whereas the n-6/n-3 ratio of SPWL fed on 7–20% PS-formulated diets was less than the recommended value of 4 (Table 6). The n-6/n-3 ratios of all present SPWL were lower than that of *Gryllus assimilis* fed with diets containing fish oil (16.07), pumpkin seed oil (55.84), and sunflower oil (54.14) [37], as well as *Allomyrina dichotoma* (39.50), *Protaetia brevitarsis* (25), *Zophobas mori* (24.80), and *Tenebri molitor* (69.72) [34]. The IA and IT indexes determine the atherogenic and thrombogenic capabilities of FA based on their proclivity to form clots in blood arteries, resulting in lower total cholesterol and LDL-cholesterol levels in blood plasma through the consumption of foods with lower IA and IT levels [36]. All SPWL had IA and IT indexes less than one (Table 6), with SPWL raised on a PI-added diet having the lowest IA value and SPWL fed only PI or PI with PS having the lowest IT value (*p* < 0.05). By significantly altering the FA composition of SPWL, it is possible to obtain healthier lipids for hypercholesterolemic or diabetic people by feeding SPWL with 7–20% PS-formulated diets.

### 3.5. FA Metabolic Indices and Δ6 Desaturase (fads2) Gene Expression

Figure 2 depicts the estimated indices of FA metabolism and Δ6 desaturase (*fads2*) gene expression. Animals do not synthesize essential LA and ALA from acetyl-CoA via -oxidation; nevertheless, desaturating, elongating, and terminating enzymes can convert dietary LA and ALA to longer chain PUFA [18]. The SPWL fed all diets demonstrated a high level of estimated Δ9-desaturase activity when converting stearic acid to oleic acid, with only a slight difference between diet groups (Figure 2A). SPWL fed a 7% PS added diet had the highest Δ9-desaturase activity in converting palmitic acid to palmitoleic acid, but SPWL fed a PI-based diet had the highest total Δ9-desaturase index (*p* < 0.05, Figure 2A). This could be attributed to the presence of the most abundant MUFA, palmitoleic acid (C16:1) and oleic acid (C18:1), in the resulting SPWL (Table 6). It should be noted that the higher level of oleic acid in SPWL may also be due to direct bioaccumulation from the diet (Table 3). Despite this, the presence of palmitoleic acid in SPWL was largely associated with Δ9-desaturase activity because this FA was naturally absent in the test diets (Table 3). Previous research in *Hermetia illucens* suggested that this 9-desaturase gene in SPWL encodes a transmembrane FA desaturase capable of synthesizing palmitoleic acid or oleic acid from corresponding SFA precursors [38]. The FA synthase complex was primarily responsible for the direct and overwhelming synthesis of palmitic acid from acetyl-CoA via malonyl-CoA synthesis, which was caused by the high affinity of the terminal thioesterase for palmitic acid [39]. The biosynthesized palmitic acid is a precursor for the formation of longer chain FA such as stearic acid (C18:0) via the FA elongase 6/ELOVL 6, which can then be desaturated to form UFA or used to synthesize storage lipids [38]. The SPWL raised on all experimental diets were slightly low in EI, ranging from 0.03–0.06 (Figure 2A). This was clearly explained by the small changes in very long-chain FA in the resulting SPWL compared to the FA in the test diets. The highest TI activity was observed in SPWL raised on a control diet (*p* < 0.05), with decreasing TI in SPWL when formulating supplements into the diets (Figure 2A). This finding was consistent with the FA profile of SPWL (Table 6), which could be due to higher levels of longer chain FA in the supplemental diets reducing TI activity in SPWL during rearing. The ability of an animal to synthesize long-chain PUFA from LA and ALA is also determined by the activity of the rate-limiting enzymes, Δ5 + Δ6-desaturases, which varies depending on substrate specificity and catalytic activity between genus and species [18]. The highest Δ5 + Δ6-desaturase activity was observed in SPWL reared on a control diet (*p* < 0.05), and it was dramatically reduced after SPWL were fed all supplemental based diets (*p* < 0.05; Figure 2A). This may contribute to a higher concentration of LA and ALA in supplemental diet groups, resulting in competition for binding to the active sites of Δ5 + Δ6-desaturase [40]. The mRNA expression pattern of *fads2* in SPWL was similar to that found in the Δ5 + Δ6-desaturase index (Figure 2B), where the highest expression of the *fads2* gene was noticeable in SPWL fed with a control diet (*p* < 0.05). The addition of PI to a regular diet reduced *fads2* gene expression to a second order; however, a reduction in these values in SPWL was observed after varying amounts of PS were added to the diets (Figure 2B). Increasing the amount of PS added to the test diets from 7 to 15% led to an increase in *fads2* gene expression of SPWL, followed by a decrease when adding 20% PS to the diet (*p* < 0.05). All SPWL fed PS-formulated diets had higher *fads2* gene expression than PF-added diet-raised SPWL (*p* < 0.05). In comparison to the relationship between longer chain PUFA and *fads2* gene expression in SPWL, a small change in ARA, EPA, and DHA content was observed after rearing SPWL with PS-formulated diets (Table 6), which was well related to the low *fads2* gene expression (Figure 2B). Despite the fact that SPWL can enzymatically synthesize n-3 longer chain PUFA from ALA, as evidenced by the presence of estimated Δ5 + Δ6-desaturase activity and *fads2* gene expression (Figure 1 and Figure 2A), the extent of all n-3 longer chain PUFA, particularly EPA, was low in this study (Table 6). Due to a lack of desaturase/elongase activity, the SPWL may only synthesize EPA de novo from ALA in small amounts through specific tissues. This result was consistent with the findings of Mattioli et al. [41], who found that *Tenebrio molitor* larvae were nearly unable to produce long-chain PUFA when fed high levels of ALA-containing diets. The larvae simply bioaccumulated ALA and converted roughly two-thirds of it to SFA, most likely lauric acid or myristic acid [36]. The current study first addressed the relationship between n-3 long-chain PUFA extent, desaturation/elongation enzyme systems, and *fads2* gene expression as altered by high ALA-containing diet feeding in SPWL, which still has some gaps to fill in order to fully understand their lipid metabolism.

### 3.6. Amino Acid Composition

Table 7 shows the amino acid profile of the SPWL as influenced by feeding composition. The highest EAA in the SPWL was lysine, which was increased after adding PI and PS into the diets (*p* < 0.05). There was a non-significant difference in lysine content between SPWL reared on PI and different PS contents in diets, which was greater than that raised on the control and PF added diets by 1.09–1.34 fold and 1.71–2.11 fold, respectively. The second most prevalent EAA in the SPWL was leucine, which was followed by valine, isoleucine, histidine, threonine, phenylalanine, methionine, and tryptophan (Table 7). Fortification of PI and PI with PS into the experimental diets increased all EAA in the resulting SPWL except threonine, indicating that mixed plant-based ingredients and PS were a rich source of EAA. It should be noted that the high concentration of PS inclusion in the diets, ranging from 15–20%, resulted in a reduction in the individual EAA counterpart in SPWL. This could be due to an imbalance in insect metabolism as a result of feeding high lipid-containing PS-added diets. SPWL contained tryptophan in concentrations ranging from 0.1 to 0.2 mg/g, a typical limited amino acid in insects [42,43]. Supplemental diets increased tryptophan levels in SPWL by approximately twofold when compared to control diets (*p* < 0.05). The total EAA of the SPWL ranged from 77.8 to 121.7 mg/g (Table 7). The highest total EAA content was found in SPWL raised on 3–7% PS added diets (*p* < 0.05). The total EAA content of SPWL was similar to soybean, which was superior to other vegetable proteins but inferior to livestock proteins [44]. However, the EAA content of the SPWL (except when raised on PF diet) was greater than that of *Rhynchophorus bilineatus* (40 mg/g) and *Cladomorphus phyllinum* (83.2 mg/g), as reported by Köhler et al. [45] and Botton et al. [42]. It should be noted that SPWL raised on PI and PI with PS had higher total EAA content than the control and PF-added diets (*p* < 0.05), suggesting that PI and PS are rich sources of EAA [5].

The SPWL raised on a PF-added diet, as well as diets containing 3%, 7%, and 20% PS, had the highest geometrical mean EAAI (Table 7), which measures protein content in relation to a highly nutritive reference egg protein [46]. Normally, the EAAI of the current SPWL was in the range of 44.5–54.2% (Table 7). The inclusion of all supplements into the regular GSPT diet resulted in an increase in the EAAI of the corresponding SPWL due to the high EAA content of those ingredients, which was consistent with our previous study on the same insect [5]. The highest BV in SPWL reared on a PF-added diet, as well as diets containing 3%, 7%, and 20% PS, was observed to be 1.2–1.3 times greater than that reared on a control diet (*p* < 0.05; Table 7). A proper balance of EAA and non-essential amino acids (non-EAA), as well as other nitrogen-containing molecules, is required for optimal BV of protein [23]. SPWL were a good source of hydrophobic amino acids (phenylalanine, alanine, leucine, isoleucine, valine, and proline), which have several bioactivities in lowering the risk of some diseases [2]. SPWL also contained 1.4–2.0 mg/g of sulphur-containing amino acids (such as methionine), which have a strong antioxidant effect by scavenging reactive oxygen species (ROS), preventing cell damage, cancer, and cardiovascular disease, and improving immune function [38]. SPWL raised on 3–7% PS diets had the greatest total non-EAA content (*p* < 0.05), with the derived amino acid, ornithine, being the most prevalent non-EAA in this insect (Table 7). As a result, this diet formula containing 3–7% PS yielded the highest total AA content of SPWL, ranging from 241.1–254.6 mg/g, and could potentially be used to improve AA content in this insect.

All EAA presented in SPWL met the FAO recommended level [47], with the exception of methionine, tryptophan, and phenylalanine, which are the limiting EAA with the lowest EAA score in SPWL (Table 8). In comparison to the reference protein, this result indicated that SPWL was a promising source of high-quality protein. The SPWL fed a 7% PS-diet had relatively high valine, leucine, isoleucine, histidine, and methionine + cysteine scores than that raised on a commercial PF-added diet (*p* < 0.05). However, there was no difference in threonine, lysine, phenylalanine, or total EEA score among SPWL raised on all experimental diets (*p* > 0.05). Interestingly, all SPWL had a very high lysine score of 4.15–5.28 fold when compared to the reference protein, which was independent of the diet formulations (*p* < 0.05). Our findings indicated that adding PI and PS to a regular diet could improve the protein quality of SPWL, which was 1.56–1.81 fold higher than the FAO reference protein [47]. As indicated by protein quality, rearing SPWL with a proper plant-based ingredient formulation could completely replace commercial animal feed. This strategy has the potential to boost commercial competitiveness in the Muslim market while also ensuring long-term sustainability.

## 4. Conclusions

The 7% PS supplementation in the mixed PI/GSPT diet resulted in positive growth performance and nutritive quality of SPWL. The enrichment diet with 7–20% PS effectively increased ALA content, resulting in a desirable n-6/n-3 ratio in the subsequent SPWL. Dietary PS formulation raised the Δ9-DI (18), total Δ9-DI, and Δ5 + Δ6 DI activities in SPWL, which was linked to an increase in FA unsaturation and PUFAs/SFAs ratio while reducing the n-6/n-3 ratio, AI, and TI of SPWL lipids. The higher dietary PS content formulated in the diets reduced the *fads2* gene expression of the resulting SPWL. When compared to the reference protein reported by FAO, the SPWL raised on 7% PS-added diet had superior EAA with high BV. Overall, SPWL fed 7% PS-formulated into a mixed PI/GSPT diet demonstrated superior nutritional quality, particularly lipids and proteins, as well as proper growth performance. This diet formulation rendered a well-balanced nutritional composition in SPWL which is satisfactory for human requirements. As a result, the proper dietary PS formulation resulted in SPWL with an excellent source of nutrients for food security and sustainability, as well as a less expensive food source that is easily accessible and affordable to locals. However, investigations should pay attention to other concerns including pesticide residues, polycyclic aromatic hydrocarbon (PAH) contamination, biogenic amine production, and microbiological quality in order to comply with safety requirements.

## Figures and Tables

**Figure 1 foods-11-02036-f001:**
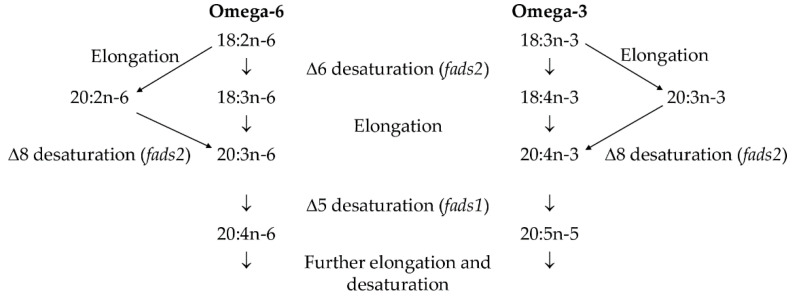
The reaction of Δ6 desaturase enzyme, encoded by the *fads2* gene, to convert the PUFA precursors to their respective metabolites.

**Figure 2 foods-11-02036-f002:**
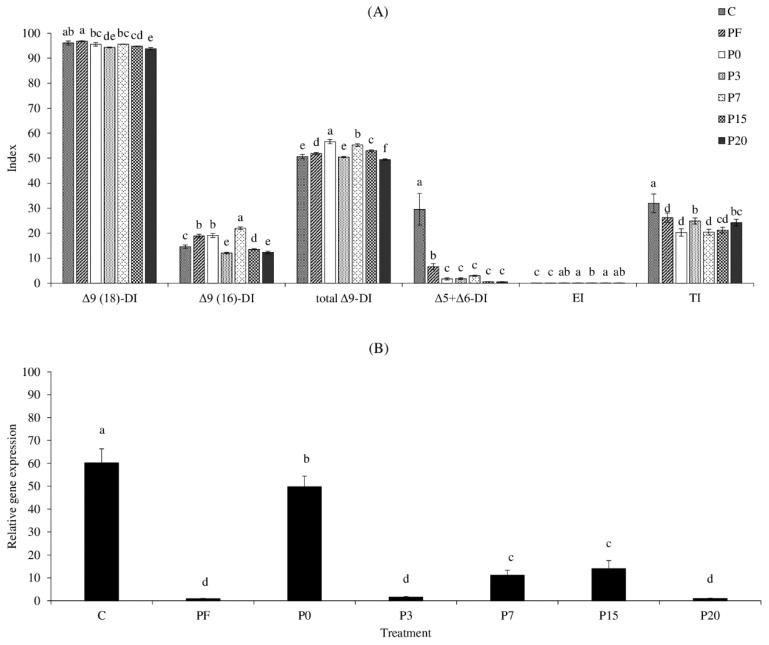
Comparison of fatty acid metabolism indices (**A**) and *fads2* gene expression (**B**) of sago palm weevil larvae (SPWL) fed on different diets. Values are given as mean ± standard deviation from triplicate determinations. Different letters in the same attribute indicate significant differences (*p* < 0.05). Experimental diet nomenclature: C: control, PF: diet supplemented with pig feed, P0-P20: diets contained 0–20% perilla seed, respectively. DI = desaturase index, EI = elongase index, TI = thioesterase index.

**Table 1 foods-11-02036-t001:** Ingredients of experimental diets.

Ingredients	C	PF	P0	P3	P7	P15	P20
Ground sago palm trunk (g)	1000	667	667	667	667	667	667
Pig feed (g)	-	333	-	-	-	-	-
Rice bran (g)	-	-	166.5	151.5	131.5	91.5	66.5
Soybean (g)	-	-	166.5	151.5	131.5	91.5	66.5
Perilla seed (g)	-	-	0	30	70	150	200

**Table 2 foods-11-02036-t002:** Proximate composition of experimental diets.

Proximate Composition	C	PF	P0	P3	P7	P15	P20
Moisture (g/100 g)	68.9 ± 0.8a	51.9 ± 2.3e	58.1 ± 0.1b	57.1 ± 0.1b	54.9 ± 0.3c	52.9 ± 0.8de	53.9 ± 0.9cd
Protein (g/100 g, dw)	1.1 ± 0.1c	10.8 ± 0.5b	10.4 ± 0.5b	10.7 ± 0.3b	11.7 ± 0.3a	10.9 ± 0.1b	11.7 ± 0.1a
Lipid (g/100 g, dw)	0.9 ± 0.2e	2.0 ± 0.0e	5.7 ± 1.2d	6.8 ± 0.8d	12.8 ± 0.4c	18.3 ± 2.3b	20.2 ± 0.5a
Ash (g/100 g, dw)	4.1 ± 0.0e	8.2 ± 0.2a	5.9 ± 0.1c	6.7 ± 0.9b	5.3 ± 0.1cd	4.7 ± 0.0de	4.4 ± 0.1e
Carbohydrate (g/100 g, dw)	93.9 ± 0.2a	79.1 ± 0.5b	78.0 ± 1.5bc	75.8 ± 1.8c	70.2 ± 0.8d	66.2 ± 2.4e	63.7 ± 0.4f

Values are given as mean ± standard deviation from triplicate determinations. Different letters in the same row indicate significant differences (*p* < 0.05). Diet nomenclature: C: control, PF: diet supplemented with pig feed, P0–P20: diets contained 0–20% perilla seed, respectively. dw = dry weight.

**Table 3 foods-11-02036-t003:** Fatty acid composition of experimental diets.

Fatty Acid (% Total Lipid)	C	PF	P0	P3	P7	P15	P20
C12:0	1.27 ± 0.27a	0.35 ± 0.02b	0.27 ± 0.06b	0.31 ± 0.05b	0.25 ± 0.06b	0.24 ± 0.02b	0.25 ± 0.03b
C14:0	0.86 ± 0.13b	5.11 ± 0.17a	0.12 ± 0.01c	0.12 ± 0.01c	0.14 ± 0.05c	0.11 ± 0.04c	0.11 ± 0.03c
C14:1	1.97 ± 0.17a	1.86 ± 0.07a	0.63 ± 0.02b	0.53 ± 0.03bc	0.43 ± 0.04c	0.40 ± 0.06c	0.39 ± 0.04c
C15:0	0.44 ± 0.02a	0.11 ± 0.02b	0.08 ± 0.02c	0.08 ± 0.01c	0.07 ± 0.01c	0.07 ± 0.01c	0.05 ± 0.02c
C15:1 (cis-10)	0.36 ± 0.25a	0.18 ± 0.04a	0.13 ± 0.02a	0.19 ± 0.05a	0.16 ± 0.05a	0.14 ± 0.06a	0.12 ± 0.03a
C16:0	34.16 ± 0.8a	21.92 ± 0.13c	23.20 ± 0.16b	19.19 ± 0.41d	15.89 ± 0.48e	13.17 ± 0.38f	12.65 ± 0.85f
C16:1 (cis-9)	0.82 ± 0.26a	0.42 ± 0.14b	0.28 ± 0.04bc	0.21 ± 0.05bc	0.20 ± 0.02bc	0.14 ± 0.03c	0.14 ± 0.03c
C17:0	0.50 ± 0.03a	0.12 ± 0.01b	0.09 ± 0.01c	0.08 ± 0.00c	0.07 ± 0.01c	0.07 ± 0.00c	0.08 ± 0.02c
C17:1 (cis-10)	0.23 ± 0.04a	0.02 ± 0.01b	0.01 ± 0.01b	0.02 ± 0.01b	0.02 ± 0.01b	0.04 ± 0.03b	0.03 ± 0.03b
C18:0	9.66 ± 0.58a	3.30 ± 0.08b	3.58 ± 0.11b	3.55 ± 0.08b	3.70 ± 0.08b	3.57 ± 0.01b	3.52 ± 0.15b
C18:1 (cis-9)	13.25 ± 0.63f	24.70 ± 0.3c	35.49 ± 0.38a	29.82 ± 0.19b	21.59 ± 0.46d	15.10 ± 0.23e	13.22 ± 0.21f
C18:2 (all-cis-9,12)	5.38 ± 0.49g	36.67 ± 0.36a	31.59 ± 0.54b	28.84 ± 0.44c	25.07 ± 0.23d	21.21 ± 0.23e	19.64 ± 0.63f
C18:3 (all-cis-cis-6,9,12)	0.21 ± 0.12a	0.04 ± 0.04b	0.04 ± 0.03b	0.10 ± 0.00b	0.05 ± 0.02b	0.02 ± 0.03b	0.05 ± 0.05b
C18:3 (all-cis-cis-9,12, 15)	1.81 ± 0.18e	1.38 ± 0.01e	1.64 ± 0.05e	14.59 ± 0.41d	30.38 ± 0.33c	43.83 ± 0.55b	47.42 ± 0.83a
C20:0	0.36 ± 0.09b	0.26 ± 0.04cd	0.46 ± 0.02a	0.37 ± 0.02b	0.31 ± 0.01bc	0.21 ± 0.04d	0.22 ± 0.03d
C20:3 (all-cis-8,11,14)	0.31 ±0.24a	0.04 ± 0.03b	0.04 ± 0.02b	0.03 ± 0.02b	0.04 ± 0.01b	0.04 ± 0.02b	0.02 ± 0.00b
C20:3 (all-cis-11,14,17)	0.57 ± 0.09a	0.08 ± 0.06bc	0.02 ± 0.01c	0.02 ± 0.01c	0.04 ± 0.01c	0.18 ± 0.14b	0.10 ± 0.05bc
C20:4n6	0.17 ± 0.13a	0.02 ± 0.01b	0.04 ± 0.02b	0.02 ± 0.03b	0.01 ± 0.00b	0.01 ± 0.01b	0.02 ± 0.01b
C20:5 (all -cis-5,8,11,14,17)	0.10 ± 0.01a	0.05 ± 0.04b	0.03 ± 0.02b	0.04 ± 0.01b	0.01 ± 0.01b	0.02 ± 0.02b	0.01 ± 0.01b
C22:2 (all-cis-13,16)	0.48 ± 0.01a	0.06 ± 0.04b	0.01 ± 0.01b	0.04 ± 0.03b	0.10 ± 0.11b	0.10 ± 0.10b	0.08 ± 0.04b
C22:6 (all-cis-4,7,10,13,16,19)	0.11 ± 0.07a	0.07 ± 0.04ab	0.02 ± 0.01b	0.03 ± 0.00b	0.04 ± 0.01b	0.04 ± 0.01b	0.05 ± 0.02b
Other	26.96 ± 1.69a	3.22 ± 0.07b	2.22 ± 0.36bc	1.80 ± 0.23c	1.44 ± 0.24c	1.29 ± 0.18c	1.84 ± 0.47c
∑SFA	47.26 ± 1.13a	31.18 ± 0.23b	27.81 ± 0.15c	23.69 ± 0.54d	20.43 ± 0.40e	17.44 ± 0.40f	16.87 ± 0.98f
∑MUFA	16.64 ± 0.45e	27.18 ± 0.19c	36.55 ± 0.34a	30.77 ± 0.30b	22.40 ± 0.43d	15.82 ± 0.25f	13.90 ± 0.13g
∑PUFA	8.96 ± 0.34d	38.50 ± 0.29c	37.90 ± 8.18c	48.95 ± 9.77b	60.06 ± 7.82a	66.90 ± 0.48a	66.94 ± 1.20a
∑n3-PUFA	2.60 ± 0.20f	1.58 ± 0.13e	1.71 ± 0.08f	14.69 ± 0.42d	30.48 ± 0.29c	44.08 ± 0.43b	47.58 ± 0.86a
∑n6-PUFA	6.07 ± 0.22g	36.78 ± 0.30a	31.70 ± 0.57b	29.00 ± 0.45c	25.16 ± 0.25d	21.27 ± 0.2e	19.73 ± 0.66f
n6/n3	2.34 ± 0.19c	23.35 ± 2.02a	18.60 ± 0.79b	1.98 ± 0.04cd	0.83 ± 0.01de	0.48 ± 0.01de	0.41 ± 0.01e

Values are given as mean ± standard deviation from triplicate determinations. Different letters in the same row indicate significant differences (*p* < 0.05). Experimental diet nomenclature: C: control, PF: diet supplemented with pig feed, P0–P20: diets contained 0–20% perilla seed, respectively. dw = dry weight.

**Table 4 foods-11-02036-t004:** Growth performance of sago palm weevil larvae (SPWL) fed with different diets.

Biometric Parameter	C	PF	P0	P3	P7	P15	P20
Survival rate (%)	90.8 ± 1.2ab	92.6 ± 2.3a	84.6 ± 2.6c	86.6 ± 0.2bc	85.9 ± 0.6c	85.8 ± 3.0c	83.8 ± 1.8c
Live weight (g)	2.6 ± 1.2c	5.3 ± 0.4ab	5.3 ± 0.6ab	5.8 ± 0.6a	5.5 ± 0.6a	5.4 ± 0.4ab	5.0 ± 0.3b
Dry weight (g)	0.9 ± 0.1b	1.7 ± 0.1a	1.6 ± 0.2a	1.9 ± 0.2a	1.7 ± 0.2a	1.9 ± 0.2a	1.8 ± 0.2a
Dry matter content (%)	26.3 ± 2.3c	30.5 ± 2.9b	35.5 ± 1.8a	33.1 ± 2.3ab	33.7 ± 1.8ab	35.2 ± 1.4a	37.0 ± 2.2a
Condition factor	8.1 ± 2.8d	9.7 ± 2.5c	11.3 ± 2.8abc	10.9 ± 1.7bc	12.6 ± 2.2a	10.9 ± 1.7bc	11.6 ± 2.9ab
Growth rate (g/day)	0.1 ± 0.1b	0.2 ± 0.0a	0.1 ± 0.0c	0.1 ± 0.0c	0.1 ± 0.0c	0.1 ± 0.0c	0.1 ± 0.0c
Appearance	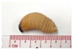	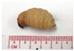	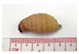	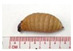	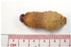	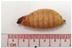	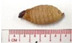

Values are given as mean ± standard deviation from triplicate determinations. Different letters in the same row indicate significant differences (*p* < 0.05). Experimental diet nomenclature: C: control, PF: diet supplemented with pig feed, P0–P20: diets contained 0–20% perilla seed, respectively.

**Table 5 foods-11-02036-t005:** Proximate composition, cholesterol content, and mineral profile of sago palm weevil larvae (SPWL) fed on different diets.

Amount (mg/g Sample)	C	PF	P0	P3	P7	P15	P20
Proximate composition							
Moisture (g/100 g)	73.7 ± 2.3a	69.5 ± 2.9b	64.5 ± 1.8c	66.9 ± 2.3bc	66.3 ± 1.8bc	64.8 ± 1.4c	62.9 ± 2.2c
Protein (g/100 g, dw)	22.3 ± 0.3c	18.6 ± 0.2d	24.5 ± 0.2b	26.7 ± 0.7a	22.6 ± 0.4c	22.7 ± 0.6c	22.1 ± 0.3c
Lipid (g/100 g, dw)	34.9 ± 0.9d	55.5 ± 2.6ab	51.1 ± 3.3c	51.0 ± 2.0c	54.6 ± 2.4abc	56.4 ± 3.0a	51.8 ± 1.0bc
Ash (g/100 g, dw)	4.2 ± 0.1a	2.8 ± 0.4b	3.8 ± 0.8ab	2.9 ± 0.1b	2.9 ± 0.1b	2.8 ± 0.2b	3.0 ± 1.2b
Carbohydrate (g/100 g, dw)	38.7 ± 0.9a	23.1 ± 2.2b	20.6 ± 3.9b	19.5 ± 2.4b	19.9 ± 2.8b	18.2 ± 3.3b	23.1 ± 1.8b
Cholesterol (mg/100 g)	246.2 ± 1.3e	350.8 ± 13.2a	288.7 ± 2.8c	303.3 ± 3.1b	261.0 ± 3.0d	297.9 ± 2.5bc	309.1 ± 12.3b
Mineral (mg/kg)							
Potassium (K)	9265 ± 286a	6487 ± 45e	7039 ± 110c	7988 ± 100b	6828 ± 137cd	6421 ± 110e	6673 ± 31de
Phosphorus (P)	3563 ± 112bc	2926 ± 55f	3639 ± 24b	4273 ± 40a	3447 ± 44d	3477 ± 64cd	3292 ± 22e
Magnesium (Mg)	1869 ± 49b	132 1± 22f	1669 ± 16c	1985 ± 10a	1631 ± 16c	1428 ± 31d	1372 ± 6e
Sodium (Na)	1351 ± 37a	986 ± 37e	1010 ± 17cd	1182 ± 11b	1035 ± 16c	1025 ± 11cd	915 ± 29d
Calcium (Ca)	699 ± 21a	441 ± 18f	625 ± 12b	674 ± 4a	549 ± 5c	514 ± 8d	478 ± 7e
Zinc (Zn)	99.4 ± 2.2b	101.9 ± 1.3ab	85.3 ± 1.8c	104.6 ± 2.0a	88.3 ± 1.5c	81.8 ± 2.3d	87.2 ± 0.4c
Manganese (Mn)	49.9 ± 1.5a	13.6 ± 0.4c	12.5 ± 0.4cd	16.7 ± 0.3b	10.4 ± 0.2f	11.0 ± 0.6f	11.7 ± 0.7de
Iron (Fe)	14.2 ± 0.2d	15.4 ± 0.4c	18.8 ± 0.3a	19.0 ± 1.1a	16.8 ± 1.0b	16.2 ± 0.6bc	15.8 ± 0.1bc
Copper (Cu)	10.2 ± 0.6d	26.1 ± 0.6a	10.8 ± 0.6cd	12.3 ± 0.5b	11.5 ± 0.4bc	11.6 ± 0.4bc	12.2 ± 0.2b

Values are given as mean ± standard deviation from triplicate determinations. Different letters in the same row indicate significant differences (*p* < 0.05). dw = dry weight. Diet nomenclature: C: control, PF: diet supplemented with pig feed, P0–P20: diets contained 0–20% perilla seed, respectively.

**Table 6 foods-11-02036-t006:** Fatty acid composition of sago palm weevil larvae (SPWL) fed on different diets.

Fatty Acid (% Total Lipid)	C	PF	P0	P3	P7	P15	P20
C12:0	0.16 ± 0.03a	0.15 ± 0.04a	0.18 ± 0.04a	0.14 ± 0.02a	0.14 ± 0.03a	0.14 ± 0.00a	0.13 ± 0.01a
C14:0	1.42 ± 0.15c	1.69 ± 0.10ab	1.76 ± 0.11ab	1.68 ± 0.08ab	1.88 ± 0.12a	1.76 ± 0.11ab	1.61 ± 0.06b
C14:1	0.06 ± 0.03a	0.03 ± 0.00a	0.11 ± 0.05a	0.05 ± 0.05a	0.12 ± 0.08a	0.09 ± 0.07a	0.05 ± 0.03a
C15:0	0.06 ± 0.02a	0.07 ± 0.01a	0.09 ± 0.02a	0.07 ± 0.00a	0.08 ± 0.03a	0.09 ± 0.02a	0.08 ± 0.00a
C15:1 (cis-10)	0.23 ± 0.18a	0.26 ± 0.08a	0.14 ± 0.10a	0.42 ± 0.10a	0.27 ± 0.19a	0.47 ± 0.06a	0.25 ± 0.04a
C16:0	45.05 ± 0.27a	43.99 ± 0.40b	35.47 ± 0.60f	41.77 ± 0.16c	38.23 ± 0.34d	37.24 ± 0.33e	38.99 ± 0.82d
C16:1 (cis-9)	7.70 ± 0.40d	10.21 ± 0.40b	8.34 ± 0.36c	5.72 ± 0.15e	10.77 ± 0.36a	5.82 ± 0.17e	5.46 ± 0.15e
C17:0	0.04 ± 0.01b	0.07 ± 0.01a	0.07 ± 0.01a	0.09 ± 0.01a	0.07 ± 0.01a	0.09 ± 0.01a	0.08 ± 0.01a
C17:1 (cis-10)	0.09 ± 0.11a	0.04 ± 0.02a	0.05 ± 0.02a	0.04 ± 0.04a	0.04 ± 0.05a	0.03 ± 0.02a	0.03 ± 0.00a
C18:0	1.64 ± 0.28c	1.28 ± 0.06d	1.89 ± 0.23bc	2.37 ± 0.05a	1.80 ± 0.04c	2.12 ± 0.04ab	2.31 ± 0.17a
C18:1 (cis-9)	40.28 ± 1.49a	38.59 ± 0.54a	40.40 ± 1.63a	39.18 ± 0.39a	38.75 ± 1.08a	38.47 ± 0.50a	34.92 ± 0.52b
C18:2 (all-cis-9,12)	0.22 ± 0.06g	11.64 ± 0.08f	8.11 ± 0.29a	5.65 ± 0.14d	3.92 ± 0.08e	6.28 ± 0.08b	5.93 ± 0.14c
C18:3 (all-cis-cis -6,9,12)	0.22 ± 0.07a	0.10 ± 0.03a	0.13 ± 0.13a	0.06 ± 0.06a	0.07 ± 0.08a	0.04 ± 0.02a	0.01 ± 0.01a
C18:3 (all-cis-cis -9,12, 15)	0.08 ± 0.01e	0.17 ± 0.02e	0.83 ± 0.07d	0.99 ± 0.05d	1.90 ± 0.05c	4.97 ± 0.06b	8.40 ± 0.45a
C20:0	0.09 ± 0.05a	0.08 ± 0.04a	0.07 ± 0.04a	0.08 ± 0.02a	0.09 ± 0.02a	0.06 ± 0.01a	0.09 ± 0.00a
C20:3 (all-cis-8,11,14)	0.22 ± 0.05a	0.03 ± 0.02b	0.12 ± 0.13ab	0.03 ± 0.04b	0.06 ± 0.04b	0.04 ± 0.03b	0.04 ± 0.01b
C20:3 (all-cis-11,14,17)	0.07 ± 0.03a	0.04 ± 0.02a	0.06 ± 0.02a	0.06 ± 0.02a	0.06 ± 0.05a	0.05 ± 0.04a	0.03 ± 0.03a
C20:4n6	0.06 ± 0.01abc	0.06 ± 0.02ab	0.04 ± 0.02bc	0.04 ± 0.01c	0.07 ± 0.00a	0.01 ± 0.00d	0.01 ± 0.01d
C20:5 (all -cis-5,8,11,14,17)	0.03 ± 0.02b	0.02 ± 0.01b	0.04 ± 0.00b	0.02 ± 0.01b	0.06 ± 0.00a	0.03 ± 0.00b	0.02 ± 0.01b
C22:2 (all-cis-13,16)	0.04 ± 0.02a	0.06 ± 0.04a	0.06 ± 0.06a	0.03 ± 0.01a	0.05 ± 0.05a	0.04 ± 0.04a	0.04 ± 0.01a
C22:6 (all-cis-4,7,10,13,16,19)	0.04 ± 0.01b	0.05 ± 0.01b	0.08 ± 0.02a	0.06 ± 0.00b	0.05 ± 0.02b	0.03 ± 0.00b	0.04 ± 0.00b
Other	2.17 ± 0.60a	1.39 ± 0.33a	1.97 ± 0.54a	1.44 ± 0.04a	1.51 ± 0.19a	2.15 ± 0.18a	1.48 ± 0.56a
∑SFA	48.47 ± 0.41a	47.33 ± 0.23b	39.53 ± 0.60f	46.20 ± 0.19c	42.29 ± 0.39e	41.51 ± 0.43e	43.29 ± 0.82d
∑MUFA	48.37 ± 1.04b	49.12 ± 0.42ab	49.04 ± 1.36ab	45.42 ± 0.38c	49.96 ± 0.61a	44.87 ± 0.49c	40.71 ± 0.30d
∑PUFA	1.00 ± 0.12d	2.41 ± 0.42d	9.50 ± 0.51b	7.23 ± 0.61c	7.25 ± 1.73c	12.81 ± 2.19a	14.32 ± 0.10a
∑n3-PUFA	0.23 ± 0.04e	0.27 ± 0.02e	1.00 ± 0.08d	1.12 ± 0.04d	2.06 ± 0.11c	5.08 ± 0.04b	8.50 ± 0.45a
∑n6-PUFA	0.72 ± 0.08f	1.83 ± 0.05e	8.41 ± 0.50a	5.79 ± 0.17c	4.13 ± 0.03d	6.37 ± 0.11b	5.99 ± 0.16c
n6/n3	3.21 ± 0.73d	6.71 ± 0.52b	8.46 ± 0.23a	5.15 ± 0.04c	2.01 ± 0.13e	1.25 ± 0.03f	0.1 ± 0.02f
PUFA/SFA	0.02 ± 0.00d	0.05 ± 0.01d	0.24 ± 0.01b	0.16 ± 0.01c	0.17 ± 0.04c	0.31 ± 0.05a	0.33 ± 0.01a
Atherogenicity index (IA)	0.41 ± 0.14ab	0.54 ± 0.12a	0.32 ± 0.09b	0.53 ± 0.04a	0.38 ± 0.03ab	0.38 ± 0.01ab	0.46 ± 0.08ab
Thrombogenicity index (IT)	0.75 ± 0.25ab	0.97 ± 0.21a	0.54 ± 0.15b	0.90 ± 0.07a	0.58 ± 0.06b	0.48 ± 0.02b	0.47 ± 0.10b

Values are given as mean ± standard deviation from triplicate determinations. Different letters in the same row indicate significant differences (*p* < 0.05). Experimental diet nomenclature: C: control, PF: diet supplemented with pig feed, P0–P20: diets contained 0–20% perilla seed, respectively.

**Table 7 foods-11-02036-t007:** Amino acid composition of sago palm weevil larvae (SPWL) fed on different diets.

Amount (mg/g Sample)	C	PF	P0	P3	P7	P15	P20
Essential amino acid (EAA)							
Valine	9.2 ± 0.2c	10.5 ± 1.9bc	10.7 ± 0.6b	12.5 ± 0.3a	11.4 ± 0.2ab	10.1 ± 0.2bc	10.7 ± 0.3b
Leucine	13.0 ± 0.4d	12.8 ± 0.6d	15.3 ± 1.3bc	17.9 ± 0.6a	16.2 ± 0.1b	14.2 ± 0.5c	14.8 ± 0.4c
Isoleucine	7.5 ± 0.2e	8.0 ± 0.4de	8.7 ± 0.5c	10.0 ± 0.4a	9.4 ± 0.2b	8.2 ± 0.1cd	8.6 ± 0.3c
Threonine	4.8 ± 0.1a	6.1 ± 2.5a	5.6 ± 0.2a	6.4 ± 0.2a	5.9 ± 0.1a	5.3 ± 0.1a	5.3 ± 0.1a
Lysine	44.4 ± 1.5ab	28.4 ± 23.5b	54.6 ± 4.8a	59.9 ± 7.8a	57.4 ± 0.9a	48.6 ± 2.0a	50.0 ± 1.6a
Histidine	6.0 ± 0.1d	6.4 ± 0.4cd	6.9 ± 0.4bc	7.8 ± 0.6a	7.3 ± 0.1ab	6.5 ± 0.2cd	6.9 ± 0.2bc
Tryptophan	0.1 ± 0.0d	0.2 ± 0.0c	0.2 ± 0.0b	0.2 ± 0.0a	0.2 ± 0.0b	0.2 ± 0.0c	0.2 ± 0.0b
Phenylalanine	3.7 ± 0.1d	4.0 ± 0.4cd	4.5 ± 0.3b	5.0 ± 0.1a	4.6 ± 0.1b	4.1 ± 0.1cd	4.3 ± 0.1bc
Methionine	1.5 ± 0.0de	1.4 ± 0.1e	1.8 ± 0.1abc	2.0 ± 0.2a	1.9 ± 0.0ab	1.7 ± 0.0cd	1.7 ± 0.0bc
Total EAA	90.2 ± 2.6cd	77.8 ± 18.5d	108.2 ± 8.2ab	121.7 ± 9.2a	114.4 ± 1.3ab	98.9 ± 2.9bc	102.6 ± 3.0bc
Essential amino acid index (EAAI)	44.5 ± 1.0c	50.3 ± 5.3ab	47.8 ± 2.8bc	49.5 ± 2.1ab	54.2 ± 0.9a	47.9 ± 0.9bc	51.4 ± 1.2ab
Biological value (BV)	36.8 ± 1.1c	43.2 ± 5.8ab	40.4 ± 3.0bc	42.3 ± 2.3ab	47.4 ± 0.9a	40.5 ± 0.9bc	44.4 ± 1.3ab
Non-essential amino acid (non-EAA)							
Tyrosine	3.8 ± 0.1a	5.7 ± 3.1a	4.5 ± 0.1a	5.0 ± 0.2a	4.6 ± 0.2a	4.2 ± 0.0a	4.3 ± 0.0a
Alanine	9.4 ± 0.2a	11.1 ± 3.3a	10.7 ± 0.6a	12.3 ± 0.5a	11.3 ± 0.4a	10.1 ± 0.2a	10.3 ± 0.2a
Glycine	17.5 ± 0.4b	15.9 ± 3.4b	21.2 ± 1.4a	23.6 ± 2.3a	22.5 ± 0.5a	20.5 ± 0.4a	21.1 ± 0.5a
Serine	3.2 ± 0.1a	3.9 ± 1.1a	3.6 ± 0.2a	4.2 ± 0.1a	3.8 ± 0.1a	3.5 ± 0.1a	3.5 ± 0.0a
Aspartic acid	5.0 ± 0.2a	8.6 ± 6.1a	6.0 ± 0.2a	6.7 ± 0.3a	6.2 ± 0.3a	5.6 ± 0.1a	5.5 ± 0.1a
Hydroxyproline	0.4 ± 0.0a	0.3 ± 0.0a	0.4 ± 0.0a	0.4 ± 0.1a	0.4 ± 0.0a	0.4 ± 0.0a	0.4 ± 0.0a
Proline	13.5 ± 0.3bc	11.1 ± 5.8c	18.3 ± 1.0ab	20.0 ± 4.7a	19.7 ± 0.4a	17.3 ± 0.2ab	19.1 ± 0.4a
Glutamic acid	9.7 ± 0.2a	14.4 ± 8.2a	10.9 ± 0.6a	12.6 ± 0.7a	11.7 ± 0.4a	10.6 ± 0.2a	10.6 ± 0.1a
Arginine	10.0 ± 0.5c	9.6 ± 0.9c	11.3 ± 0.3b	12.5 ± 0.9a	10.6 ± 0.2bc	9.6 ± 0.3c	10.1 ± 0.1c
Ornithine	24.0 ± 1.1b	23.5 ± 4.3b	29.5 ± 3.4a	34.4 ± 4.4a	34.7 ± 0.4a	30.0 ± 1.6a	31.5 ± 1.1a
Glutathione	0.6 ± 0.0c	0.7 ± 0.1b	0.7 ± 0.0b	0.8 ± 0.0a	0.7 ± 0.0b	0.6 ± 0.0c	0.7 ± 0.0b
Cystine	0.4 ± 0.0c	0.4 ± 0.0bc	0.5 ± 0.0a	0.5 ± 0.0a	0.5 ± 0.0ab	0.4 ± 0.0c	0.5 ± 0.0ab
Total non-EAA	97.6 ± 2.8d	105.1 ± 9.2cd	117.6 ± 7.7bc	132.9 ± 13.0a	126.7 ± 2.4ab	112.7 ± 2.7c	117.5 ± 2.5bc
Total amino acid	187.9 ± 5.4d	183.0 ± 9.4d	225.9 ± 15.8bc	254.6 ± 22.0a	241.1 ± 3.6ab	211.6 ± 5.7c	220.0 ± 5.3bc

Values are given as mean ± standard deviation from triplicate determinations. Different letters in the same row indicate significant differences (*p* < 0.05). Diet nomenclature: C: control, PF: diet supplemented with pig feed, P0–P20: diets contained 0–20% perilla seed, respectively.

**Table 8 foods-11-02036-t008:** Recommended amino acid scores for adult (mg/g protein) of sago palm weevil larvae (SPWL) fed with different diets compared to referent protein.

Essential Amino Acid	C	PF	P0	P3	P7	P15	P20	Reference Protein ^#^
Valine	41.1 ± 0.9c (1.03) *	56.7 ± 10.1a (1.42)	43.5 ± 2.6bc (1.09)	47.0 ± 1.1bc (1.18)	50.5 ± 1.1ab (1.26)	44.6 ± 0.8bc (1.11)	48.5 ± 1.2bc (1.21)	40
Leucine	58.1 ± 1.9c (0.95)	68.8 ± 3.0a (1.13)	62.3 ± 5.3bc (1.02)	67.2 ± 2.4ab (1.10)	71.6 ± 0.6a (1.17)	62.7 ± 2.1bc (1.03)	66.9 ± 1.9ab (1.10)	61
Isoleucine	33.4 ± 0.9d (1.11)	42.8 ± 2.2a (1.43)	35.3 ± 2.0cd (1.18)	37.4 ± 1.3bc (1.25)	41.5 ± 1.0a (1.38)	36.2 ± 0.4c (1.21)	38.8 ± 1.4b (1.29)	30
Threonine	21.7 ± 0.4a (0.87)	33.1 ± 13.5a (1.32)	22.9 ± 1.0a (0.92)	24.0 ± 0.7a (0.96)	26.1 ± 0.6a (1.04)	23.2 ± 0.5a (0.93)	24.2 ± 0.4a (0.97)	25
Lysine	199.3 ± 6.7a (4.15)	225.7 ± 7.1a (4.70)	222.7 ± 19.6a (4.64)	224.7 ± 29.4a (4.68)	253.5 ± 3.9a (5.28)	214.3 ± 8.7a (4.46)	226.5 ± 4.0a (4.72)	48
Histidine	26.8 ± 0.3d (1.67)	34.3 ± 2.4a (2.15)	28.2 ± 1.7d (1.76)	29.2 ± 2.1cd (1.82)	32.4 ± 0.6ab (2.02)	28.6 ± 0.8cd (1.79)	31.2 ± 0.8bc (1.95)	16
Tryptophan	0.6 ± 0.0d (0.10)	0.8 ± 0.1a (0.13)	0.7 ± 0.0c (0.11)	0.7 ± 0.0c (0.11)	0.7 ± 0.0bc (0.11)	0.7 ± 0.0c (0.10)	0.8 ± 0.0b (0.12)	6.6
Phenylalanine + Tyrosine	33.9 ± 0.9a (0.83)	52.2 ± 19.0a (1.27)	36.9 ± 1.6a (0.90)	37.8 ± 1.4a (0.92)	40.5 ± 1.4a (0.99)	36.4 ± 0.4a (0.89)	38.6 ± 0.8a (0.94)	41
Methionine + Cysteine	6.9 ± 0.2c (0.30)	7.6 ± 0.3b (0.33)	7.4 ± 0.6bc (0.32)	7.4 ± 0.8bc (0.32)	8.4 ± 0.1a (0.37)	7.4 ± 0.1bc (0.32)	7.9 ± 0.2ab (0.34)	23
Total EAA	421.8 ± 11.9a (1.45)	522.1 ± 50.3a (1.55)	459.8 ± 33.8a (1.58)	475.5 ± 35.0a (1.64)	525.3 ± 6.6a (1.81)	454.1 ± 13.1a (1.56)	483.4 ± 13.6a (1.66)	290.6

Values are given as mean ± standard deviation from triplicate determinations. Different letters in the same row indicate significant differences (*p* < 0.05). Experimental diet nomenclature: C: control, PF: diet supplemented with pig feed, P0–P20: diets contained 0–20% perilla seed, respectively. * Numbers in the parentheses represent the score ratio for each EAA in the SPWL fed with different diets and in the reference protein (EAA score). ^#^ FAO [47].

## Data Availability

Data is contained within the article.

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
