# Peer review of "A Novel Strategy for the Production of Edible Insects: Effect of Dietary Perilla Seed Supplementation on Nutritional Composition, Growth Performance, Lipid Metabolism, and Δ6 Desaturase Gene Expression of Sago Palm Weevil (Rhynchophorus ferrugineus) Larvae"

_foods, 2022, doi:10.3390/foods11142036_

Round 1
Reviewer 1 Report
The work talks about edible insects farming and the impact of the insect diets on their nutritional value, growth performance and metabolism.
In an overall analysis the work is well structured and very informative. However, English revision is necessary.
I recommend the addition of a figure with the enzymatic reactions for fads 2 gene, in order to further improve the paper readability and interest.
Specific corrections
Line 38-40 – rewrite please
Line 50 – less than 5 what?
Line 63-65 – please add a reference to the enzymatic reactions.
Line 79 – please add some information about the fads 2 genes. What is their influence and
importance. –I know it might be further down in the test but a brief explanation in the
introduction is needed.
Line 127-128 – add a brief explanation for the difference between the conversion factor.
Line 325 - This finding was (in)line (…) please check the English through out the text.
Author Response
Reviewer#1
The work talks about edible insects farming and the impact of the insect diets on their nutritional value, growth performance and metabolism.
In an overall analysis the work is well structured and very informative. However, English revision is necessary.
Ans: English was polished by Dr. Rodney Marsh from UK.
I recommend the addition of a figure with the enzymatic reactions for fads 2 gene, in order to further improve the paper readability and interest.
Ans: Figure 1 was added. “Figure 1. The reaction of D6 desaturase enzyme, encoded by the fads2 gene, to convert the PUFA precursors to their respective metabolites.”
Specific corrections
Line 38-40 – rewrite please
Ans: English was polished by Dr. Rodney Marsh from UK.
Line 50 – less than 5 what?
Ans: It was stated that “The ratio of n-3 and n-6 polyunsaturated fatty acid (PUFA) should be less than 5 for optimal human health [7].”
Line 63-65 – please add a reference to the enzymatic reactions.
Ans: Done.
Line 79 – please add some information about the fads 2 genes. What is their influence and
importance. –I know it might be further down in the test but a brief explanation in the
introduction is needed.
Ans: The information about the fads 2 genes was added.
Line 127-128 – add a brief explanation for the difference between the conversion factor.
Ans: We described that “To avoid overestimation of protein content due to the presence of some non-protein nitrogen in SPWL, a conversion factor of 6.25 was used for the diets, but a conversion factor of 5.6 was used for the SPWL”.
Line 325 - This finding was (in)line (…) please check the English through out the text.
Ans: English was polished by Dr. Rodney Marsh from UK.

Reviewer 2 Report
The paper describes a novel strategy for the production of edible insect. The work is very interesting. However, there are some revisions, especially for the English language:
1) I recommend a revision of the English language
2) In the Introduction section the novelty and the aim of the study is missing. Furthermore, the scientific opinion of the risk profile of edible insect, the European and non-European regulations are missing.
3) Line 84-88 please rephrase the sentence in a better English.
4) Line 38-40 please rephrase the sentence in a better English.
5) Line 68-70 please rephrase the sentence in a better English.
6) Table 1. Line 103: dw and NS are not displayed in the table.
7) Line 134 (FA composition) and Line 180 (Cholesterol content) the chromatographic condition and the column used are missing.
8) Line 186 Amino acid profile: How were determinated the amino acid composition of SPWL? Please explain the method used.
9) Table 3, table 6, table 7 and table 8. If you use different letters for explain significant difference, you do not use the abbreviation ns (non significant different). For non significant difference use the same letter.
10) Line 631. In the conclusions section, for defined a food security, it is also necessary evaluate other feature, such as the pesticide residues, PAH (polycyclic aromatic hydrocarbon), etc. These are missing. 1) Please explain briefly the risk and benefit of using the edible insect.
Author Response
Reviewer#2
The paper describes a novel strategy for the production of edible insect. The work is very interesting. However, there are some revisions, especially for the English language:
- I recommend a revision of the English language
Ans: English was polished by Dr. Rodney Marsh from UK.
- In the Introduction section the novelty and the aim of the study is missing. Furthermore, the scientific opinion of the risk profile of edible insect, the European and non-European regulations are missing.
Ans: The novelty and the aim of the study were added. “The objective of this study was to investigate the effect of various PS concentrations in a formulated diet on growth performance, nutritional value, and the activities of lipid metabolism enzymes, including the expression of fads 2 genes in SPWL. This is the first study on the use of dietary supplementation from perilla seeds as a cutting-edge method for creating edible SPWL. ”
The scientific opinion of the risk profile of edible insect was given.
The use of insects as a food and feed source is thought to have major benefits for the economy, the environment, and food security. Although it is still a very small niche industry in the EU, insect farming for food and feed is growing in popularity [3]. Additionally, there is a challenge with the import of insects and insect-derived products into the EU for food and feed because the use of insects is more common outside the EU [3]. Considering the full supply chain, from farming to consumption, a risk profile and presentation of potential biological, chemical, allergenicity, and environmental concerns connected with farmed insects used as food and feed was developed [3].
- Line 84-88 please rephrase the sentence in a better English.
Ans: English was polished by Dr. Rodney Marsh from UK.
- Line 38-40 please rephrase the sentence in a better English.
Ans: English was polished by Dr. Rodney Marsh from UK.
- Line 68-70 please rephrase the sentence in a better English.
Ans: English was polished by Dr. Rodney Marsh from UK.
6) Table 1. Line 103: dw and NS are not displayed in the table.
Ans: Thank you very much. They were removed.
7) Line 134 (FA composition) and Line 180 (Cholesterol content) the chromatographic condition and the column used are missing.
Ans: We are unable to add the method's details due to a similarity index issue, however readers can discover the method in the reference list. The references from our prior paper contain information about the chromatographic setup and the column that were used to measure fatty acid and cholesterol.
8) Line 186 Amino acid profile: How were determinated the amino acid composition of SPWL? Please explain the method used.
Ans: Due to the similarity index issue, we cannot add the detail for the method of amino acid determination but readers can find the method from the reference list, which was from our publication.
9) Table 3, table 6, table 7 and table 8. If you use different letters for explain significant difference, you do not use the abbreviation ns (non significant different). For non significant difference use the same letter.
Ans: Done.
10) Line 631. In the conclusions section, for defined a food security, it is also necessary evaluate other feature, such as the pesticide residues, PAH (polycyclic aromatic hydrocarbon), etc. These are missing. 1) Please explain briefly the risk and benefit of using the edible insect.
Ans: The statement was added. “However, investigations should pay attention to other concerns including pesticide residues, polycyclic aromatic hydrocarbon (PAH) contamination, biogenic amine production, and microbiological quality in order to comply with safety requirement.”

Round 2
Reviewer 2 Report
I would like to thank the Authors for taking into account the Reviewer's recommendations.